# Inclusion of young people with disabilities in the future of work: forecasting workplace, labour market and community-based strategies through an online and accessible Delphi survey protocol

Arif Jetha ,[1,2] Kay Nasir,[1] Dwayne Van Eerd ,[1] Monique A M Gignac ,[1,2] Kathleen A Martin Ginis ,[3,4,5,6] Emile Tompa[1,7]

For numbered affiliations see end of article.

**Correspondence to**
Dr Arif Jetha; AJetha@iwh.on.ca

## ABSTRACT

**Introduction** The future of work is expected to transform the nature of work, create unique employment barriers for young people living with disabilities and disrupt pathways to better health. We present a Delphi survey protocol through which we aim to obtain future-oriented strategies that can improve the accessibility and inclusion of young people with disabilities in the future of work.

**Methods and analysis** The Delphi survey will be conducted primarily online, over two rounds and in a format that is accessible to people living with disabilities. A diverse sample of subject matter experts (eg, policy makers, employment service providers, labour market experts) and participants with lived experience of a disability will be recruited using a purposive sampling strategy. All participants will be asked to complete both rounds of the Delphi survey. In the first round, open-ended questions will be asked about workplace, community-based or policy supports that can foster the inclusion of young people with disabilities in the labour market and that can also address specific future of work trends which span sociopolitical, economic, environmental and technological domains. In the second round of the survey, we will aim to build consensus; participants will be provided with a summary of specific strategies that correspond to the different future of work trends emerging from round one and will be asked to rank-order strategies according to their importance. Following the completion of the second round, consensus-based and future-focused recommendations will be generated that can support young people with disabilities in the world of work over the coming decades.

**Ethics and dissemination** The study protocol has been cleared by the University of Toronto's research ethics board (#40727). The study will identify future-focused support strategies that will be shared with people living with disabilities, policy makers and disability employment service providers through an integrated knowledge transfer and exchange approach.

## BACKGROUND

Over the next decade, the nature and availability of work is expected to transform across

## STRENGTHS AND LIMITATIONS OF THIS STUDY

⇒ The Delphi survey elicits future-focused recommendations that can support the involvement of young people with disabilities within the future of work.
⇒ By collecting anonymous expert opinions and feedback the Delphi survey facilitates unbiased debate with the goal of reaching consensus on strategies to support the employment of young people with disabilities.
⇒ Multiple Delphi survey rounds that build on one another can establish consensus across participants on strategies to support the employment of young people with disabilities.
⇒ Generating employment support strategies from a future-facing perspective could be difficult for study participants and represents a limitation.

most sectors. These changes are predicted to exacerbate employment challenges for labour market groups who have traditionally faced disadvantages.[1] Young people living with disabilities are one specific group who face difficulties entering and advancing within the world of work and can benefit from strategies that promote inclusion.[2 3] Indeed, changes to work necessitate the redesign and innovation of policies and programmes that can be implemented within workplace, community or policy settings to support the engagement of young people with disabilities in employment.

Research on industrialised countries indicates that young people with disabilities face challenges finding and sustaining employment. Data from Canada, where the study will be conducted, show that the employment rate of youth (18–24 years) and young adults (25–35 years) living with a disability is significantly lower than their healthy counterparts (32%

and 54% compared with 52% and 82%, respectively).[2 4] Young people with disabilities who find paid employment are more likely to report fewer work hours, lower income, lost productivity and barriers to career advancement than their non-disabled peers.[4 5] The barriers to transitioning to employment reported by young Canadians with disabilities parallel documented studies conducted in other countries such as Australia, Sweden and the United Kingdom.[6–8] Of concern, challenges faced by young people with disabilities during the early career phase can have a lasting effect and contribute to adverse labour market outcomes that extend across the working life course.[9]

Improving accessibility and inclusion for young people with disabilities in paid employment represents a priority for employers and social policy policy makers and has significant implications for public health. Strategies that address accessibility in the physical work environment, strengthen interpersonal relationships, address discrimination, encourage job autonomy and facilitate healthcare access are important to supporting the employment of people with disabilities.[5 10–12] The absence of these strategies can contribute to missed workdays, reduced work hours and greater workplace activity limitations. To date, research on the accessible and inclusive employment of young people with disabilities entering and advancing in the world of work have been developed within existing labour market contexts and may not account for potential changes to work that are expected to occur in the future.

It is anticipated that in the future, work will be characterised by interrelated social, technological, environmental, economic and political driving forces that are expected to disrupt every industry, transform working conditions and affect the types and availability of jobs.[1 13] The rapid digital transformation of the economy is considered a defining feature of the future of work. The development and application of diverse digital technologies—such as artificial intelligence (AI) and machine learning systems, advanced robots, smart sensors, the Internet of Things (IoT), blockchain technology or self-driving vehicles—are occurring at a faster pace compared with past periods of technological adoption, resulting in significant change to the world of work.[14–17] Of note, the COVID-19 pandemic has accelerated employer investment into AI applications and other digital technologies to maintain productivity amidst lockdown measures and to address safety concerns related to disease transmission.[13 18] A recent synthesis of literature on the future of work noted that it will also be characterised by social (eg, generational changes in the workforce), political (eg, growing populist sentiments) and environmental changes (eg, climate change) that can create opportunities and challenges for workers.[1]

Changes to work have the potential to contribute to productivity gains, create new occupations and enable new working arrangements in ways that can increase employment opportunities and demand for diverse workers.[19] At the same time, studies highlight that the future of work could be marked by widening labour market inequity such as a more rapid erosion of standard employment opportunities (eg, full-time permanent jobs) and growth of non-traditional employment (eg, freelancing, gig work) coupled with wage stagnation.[19] Of concern, disadvantaged groups of workers, including people living with disabilities, may be particularly susceptible to disadvantage in the future of work.[1 16 20] Other studies indicate that the growing application of advanced technologies like machine learning within workplaces can enable the automation of increasingly complex and predictive job tasks.[16] Accordingly, advanced technologies have the potential to contribute to the displacement of a broader subset of workers including those with greater educational attainment and who are employed in higher-skilled and professional occupations.

Inclusive strategies are required for young people with disabilities within the future of work that provide protection from potential disruptions and can increase access to job opportunities. Drawing on a social determinants of health framework, the integration of young people with disabilities in the future of work has the potential to strengthen longer-term socioeconomic pathways to better health and quality of life.[21 22] Moreover, promoting the employment of people with disabilities has broader societal (eg, decreased demand for income support or social assistance) as well as commercial benefits (eg, greater access to a highly skilled and motivated workforce to address labour shortages).[23] There is a need to design strategies that can ensure that the most vulnerable workers are protected from adverse changes in the future of work and are able to access potential opportunities. It is important to describe and implement a study protocol that can collect insights of diverse experts to forecast strategies that can be implemented in the present and ensure sustained employment of young people living with disabilities in the future.

We describe an application of an online accessible Delphi survey protocol that aims to collect insights from a range of experts on the strategies that can support the involvement of young people with disabilities within the future of work. Our planned study has three specific objectives:

1. Identify and build consensus on the specific workplace supports, policies or community-based programmes for young people with disabilities that can foster inclusion in the future of work.
2. Determine the barriers that young people with disabilities could encounter to accessing workplace supports, policies or community-based programmes within the context of work in the future.
3. Design concrete recommendations for employers, policy makers and community-based organisations to promote inclusion of young people with disabilities in the future of work.

## METHODS AND ANALYSIS
### Context
Sociopolitical and economic contextual factors play a significant role in determining the challenges and

opportunities in the future of work as well as the types of labour market policies or employment programmes that can be implemented to support young people living with disabilities.[13] Our study will be implemented in Canada, where the federal government recently passed the Accessible Canada Act (ACA), legislation that identifies employment as one of seven priority areas. The ACA legislates the creation of a barrier-free Canada through proactive identification, prevention and removal of barriers to accessibility in federally regulated workplaces.[24] Passing the ACA has moved Canada closer to meeting its human rights and equity obligations under the United Nations Convention on the Rights of Persons with Disabilities (CRPD) to which it is a signatory.[25] To successfully ensure accessibility within work environments for people with disabilities and achieve the legislative goals of the ACA, strategic planning is required to anticipate potential barriers that could emerge within the context of the changing nature of work. Of note, findings from our study have implications for other industrialised countries, especially those who are also signatories to the CRPD.[25]

Our Delphi survey emerges from a large-scale evidence synthesis which found that the future of work can cwiden employment inequities.[1] The synthesis uncovered nine key trends characterising the future of work: the digital transformation of the economy, AI and machine learning-enhanced automation, AI-enabled human resource management systems, skill requirements for the future of work, Globalization 4.0, climate change and the green economy, GenZs and inclusive work environments, populism and the future of work and external shocks that accelerate the changing nature of work. The study also found that these trends were interrelated and could individually and collectively contribute to barriers and opportunities for young people with disabilities in the working world. The Delphi survey we describe will enable participants to generate strategies that specifically address the nine future of work trends that were identified in the recent evidence synthesis.[1]

### Conceptual approach and Delphi survey overview

Our protocol draws on methods from the field of strategic foresight to obtain future-oriented strategies. One approach commonly used in strategic foresight is a Delphi survey to obtain insights on uncertain and complex phenomena and develop related solutions. The Delphi survey is a participatory process which draws on the knowledge of diverse experts.[26 27] The Delphi survey takes a structured and multistaged approach to collect anonymous expert opinions and feedback, facilitate unbiased debate and reach consensus on a phenomenon with the ultimate goal of generating future-oriented recommendations that contribute to policy and programmatic development.[27] The Delphi survey was initially developed by the RAND Corporation in the 1950s for technological forecasting in US military research applications[27] but has been increasingly applied to policy and programmatic development in a range of sectors including technology,

finance, education and health.[28–31] More recently, online Delphi surveys have become more common and enable participants to complete the study at their own time and pace and allow for researchers to collect insights from across different geographical locations. An online Delphi survey format also ensures the anonymity of participants and can limit biases that emerge in traditional forms of in-person group decision-making. Furthermore, an online format can ensure that data collection is accessible to participants with disabilities and compatible with assistive devices (eg, completion of Delphi survey using screen readers).[28 32–35]

### Sample recruitment

A key element in the design and implementation of the Delphi survey in this study is the selection of Canadian and international participants with subject matter expertise on the barriers and facilitators to the employment of young people with disabilities or the changing world of work.[36] Subject matter experts recruited for this study will include policy makers, disability employment service providers, educators, clinicians, futurists and young people with the lived experience of a disability. All eligible participants should be above 18 years of age and willing to participate in an open-ended survey where they can provide suggestions on the design of innovative strategies to support the employment of young people with disabilities within the context of different future of work trends.

Recruitment will be purposive to ensure we capture a wide range of expertise. Participants will be recruited with the support of our advisory panel (consisting of young people with the lived experience of a disability and representatives from national disability organisations) and research colleagues who will send direct study invitations to eligible subject matter experts and people with lived experience of a disability. Additionally, recruitment information will be shared via social media and listservs maintained by the national disability organisations to which our advisory panel are representatives. We will take a snowball approach to recruitment. Participants who complete the Delphi survey will be asked to nominate other potential participants who may be contacted. Currently, no established guidelines exist on the number of participants required for a Delphi survey. Researchers using a Delphi approach suggest that sample sizes should be large enough to capture breadth and depth in perspectives but also be manageable for the research team to enable retention across multiple survey rounds. Aligning with past Delphi studies of a similar nature to our protocol,[37] we aim to recruit up to 200 participants to enable descriptive analyses required to establish consensus and address potential challenges related to following up with participants.[38]

### Survey design

The Delphi survey will be implemented in at least two rounds in English or French languages. According to past applications of Delphi surveys, two rounds can be optimal

limit participant fatigue and dropout which may bias results.[39] The survey was developed based on multiple stages of pilot testing with members of our advisory panel. Pilot testing ensured face validity of questions, the use of comprehensible language and that the survey could be completed within 20 minutes. As well, pilot testing ensured that the format was inclusive for participants with different disabilities and could be compatible with assistive devices.

Delphi survey items were developed to foster virtual brainstorming and debate and to capture participant insights on specific strategies that can support the employment of young people with disabilities and are relevant to different trends in the future of work. Prompts are included throughout the survey to encourage participants to think strategically about the future and consider how changing conditions could impact young people with disabilities and their need for workplace, community-based and policy supports. The first survey round will be conducted between June and October 2021 and focuses specifically on idea generation and obtaining specific strategies that can support entry and advancement of young people with disabilities in the future of work.[1] Participants will be presented with a description of a trend and provided with additional readings where they can learn more on the topic if required. A series of questions corresponding to the specific trend will be asked. Participants will be asked to think about the importance of the trend to shaping the long-term employment of people with disabilities. Also, participants will be asked about whether strategies can be implemented within the community, workplace or at the policy-level to address the impact of the trend on the employment of young people with disabilities. Two open-ended questions will ask participants to describe specific strategies to address the impact of the trend, and to explain the rationale behind their responses. Overall, participants will complete this process for four different trends to which they will be randomly allocated. We also include a final open-ended question for participants to describe any additional reflections on the future of work and its implications for young people living with disabilities.

All open-ended questions from the first round of the Delphi survey will be coded using thematic analysis and an iterative approach. Specifically, we will take a constructivist perspective to inductively examine themes that emerge from the data to build an understanding of the specific strategies that can promote employment engagement in the future of work for young adults with disabilities.[40 41] To implement our analytical approach, open-ended survey responses will be reviewed by members of the research team, and a codebook will be developed through several conversations. The codebook will be applied to analyse Delphi survey responses. A primary and secondary coder will conduct a line-by-line coding of responses. Throughout the analytical process, codes and themes emerging from the Delphi survey will be discussed in research team analysis meetings and any

inconsistencies will be resolved by consensus. Through our analytical process, a standardised list of strategies that are relevant to each trend in the future of work will be generated. A summary of findings will be integrated into a short report. Findings will be member-checked by our advisory panel to ensure trustworthiness and applicability. The study team will ensure that findings are presented anonymously so that participant responses are not biased in subsequent phases of the Delphi survey.[42]

The second round of the Delphi survey, administered approximately 6 months after the first survey (May–October 2022), aims to reach consensus across study participants. Before completing the second survey, participants will be asked to read a brief report summarising the findings from the previous round. The report will serve as a resource for participants who will will be asked to complete the second round of the survey while considering findings from the first round. Participants will be presented with the trends that they were presented with in round one and the corresponding strategies that were generated. They will be asked to rank order the specific strategies that have been proposed according to their importance in supporting the employment of young people living with disabilities in the future of work. Additionally, an open-ended question will be posed to participants to ask about any additional perspectives on the future of work that could have emerged after the first round of the survey.

The results of the second Delphi survey will be summarised using descriptive statistics.[27] There exists no widely used recommendations on cut-off points to establish consensus in a Delphi survey. Aligning with past applications of the Delphi survey approach, consensus will be reached if at least 70% of participants report agreement on the ability of specific strategies to address a future of work trend.[37] We anticipate that there may be future of work strategies which do not achieve the threshold for consensus but may still be effective in promoting employment participation. We will document discordant strategies and share them in our final publications to encourage future research on their effectiveness. Open-ended questions will be thematically coded using the same iterative approach that is taken in the analysis of findings in Delphi survey round one. Following the completion of the second round of the Delphi survey, we anticipate having generated consensus-based policy and programmatic recommendations that can support young people with disabilities that are relevant to the future of work.

## Patient and public involvement

Our study takes an integrated knowledge transfer and exchange approach (iKTE) which is defined as a process of two-way exchange between researchers and diverse stakeholders across all phases of the research process including study design and implementation, analysis and synthesis and dissemination to increase research relevancy and evidence uptake and application.[43–45] To

implement an iKTE approach in our protocol, we established an advisory panel consisting of young people with lived experience of a disability[45] and representatives from national non-profit disability organisations. Advisory panel members have been engaged to inform study design, recruitment and accessibility of data collection approaches. The advisory panel will be asked to help interpret findings that emerge from the first and second rounds of the Delphi survey. Advisory panel members will also be engaged in the cocreation of study recommendations and knowledge mobilisation outputs.

## Ethics and dissemination

The research has been cleared by the University of Toronto's Research Ethics Board (#40727). All data collection and analysis will be performed in accordance with research ethics guidelines and regulations for research on human subjects. Of note, all participants will be presented with detailed study material and informed consent will be obtained prior to their participation in the study. Honorariums will be provided to participants with lived experience of a disability. All data we collect will be anonymised and stored on a secure server. Our iKTE approach will produce findings that are relevant to the unique lived experience of young people with disabilities and useable to stakeholders to inform strategies that support employment in the future of work.[44] In collaboration with our advisory panel, we will design outputs that are tailored to and shared with diverse knowledge users including policy makers, disability employment service providers, employers and people living with disabilities.

## DISCUSSION

Young people with disabilities represent a vulnerable labour market subgroup who may require strategies to address the challenges associated with entering and advancing within the world of work. In the future, the nature and availability of work is expected to shift and, as a result, create new barriers and facilitators to employment participation. The Delphi survey methodology that we present has the potential to contribute to future-oriented insights regarding the specific strategies that can be used by employers, policy makers and employment service providers to promote the employment of people with disabilities that are resilient to the changing nature of work. Importantly, the strategies produced through the Delphi survey can strengthen present and future socioeconomic pathways to health and quality of life of young people with disabilities.[46]

Past studies of young people with disabilities highlight the importance of diverse support strategies to foster employment and career advancement including workplace practices (eg, job accommodation), labour market policies (eg, antidiscrimination laws) and community-based programmes (eg, work-integrated learning).[3][12] However, social, political, technological and environmental changes in the future of work will transform workplace and labour market conditions.[1][17] As a result, existing support strategies for people with disabilities will require innovation. In this paper, we describe a Delphi survey which is a well-suited methodology to undertake exploratory research and to develop insights on the disability support strategies in the future of work. Findings from our study can be used to update existing policies and programmes for people with disabilities or inform the design of innovative initiatives that are specifically relevant to the future of work. Past scholars note that Delphi surveys have the advantage of being a rapid approach to collecting and combining expertise from diverse subject matter experts across geographic regions.[37] A Delphi approach may not replace methods which establish causation. Intervention research will be required to test the support strategies uncovered in the Delphi study in terms of their ability to foster the long-term employment of young people with disabilities.

A key objective of this Delphi survey will be to facilitate the convergence of future-focused perspectives among participants and forecast support strategies.[47] By administering the survey over two rounds, participants will be exposed to the responses of others. In doing so, they will explore their underlying assumptions regarding the future of work which may have informed their responses and compare their perspectives to those held by other subject matter experts.[27] While the focus of our Delphi approach is to establish consensus on future of work support strategies, we will also review strategies that have been proposed where there is a lack of consensus.[39] Additional follow-up research may be required to examine the potential implications of strategies that are recommended by participants and where consensus is not reached to determine their utility towards supporting young people with disabilities

Research and practice on the employment of people with disabilities highlight the necessity of multiple perspectives in identifying different barriers and facilitators to participation in paid work and develop corresponding solutions.[48] We will recruit policy makers from different levels of government as well as labour market specialists and employment service providers from across Canada to capture a range of insights regarding the strategies that correspond to changes in the future of work. Drawing on a diversity of insights can contribute to the development of future-focused wraparound supports which are seen as an important strategy to foster the employment of people with disabilities. Moreover, disability scholars highlight an imperative of ensuring that lived experience predominates research on people with disabilities.[48][49] Our study takes an iKTE approach; we created an advisory panel where we integrate lived experience into study design and implementation, data analysis and knowledge mobilisation. Through our iKTE approach, we will produce findings that are applicable to diverse knowledge users and can be implemented to address different challenges faced by young people with disabilities in the future of work.[44][45] We anticipate that our findings will also have

indirect outcomes for advisory panel members and study participants. By participating in the Delphi survey, participants will be encouraged to take a future-focused perspective towards understanding the employment challenges and opportunities faced by young people with disabilities and take a more proactive approach to designing support strategies.[50] It is important to note that while our study is conducted within the context of Canada's labour market and social policy context, it will produce insights that are relevant for the inclusion of people living with disabilities in other industrialised countries, especially those that are signatories of the CRPD.

There are some limitations of our Delphi survey. First, for subject matter experts, thinking about work-related challenges and opportunities for young people with disabilities and generating support strategies from a future-facing perspective could be difficult. To address this challenge, we have included prompts throughout the survey that encourage participants to consider the relevancy of their responses for the future. Second, our purposive sampling strategy used identify diverse subject matter experts and people with lived experience of a disability could be another study limitation. While representativeness of the sample is not a primary goal, we will take steps to ensure depth and breadth in survey responses and in the development of recommendations to support young people with disabilities in the future of work.[42] Third, loss to follow-up represents a barrier to establishing consensus and could be a source of bias. Although time and resource intensive, members of the research team will take an active approach to engaging and following up with all participants in the second phase of the Delphi survey to address the potential for dropout.

For young people with disabilities, employment represents a critical social determinant of health. Yet, many young people report challenges as they enter and advance within the working world. In this protocol, we describe a Delphi survey study protocol that will generate recommendations and build consensus on future-focused employment support strategies drawn from a diverse group of subject matter experts and people with lived experience of a disability. The evidence to be generated from our study has the potential to be used to inform the design of workplace, community-based and policy-level strategies to ensure that work in the future is inclusive of young people with disabilities.

**Author affiliations**
<sup>1</sup>Institute for Work and Health, Toronto, Ontario, Canada
<sup>2</sup>Dalla Lana School of Public Health, University of Toronto, Toronto, Ontario, Canada
<sup>3</sup>Department of Medicine, Division of Physical Medicine & Rehabilitation, University of British Columbia, Vancouver, British Columbia, Canada
<sup>4</sup>Centre for Chronic Disease Prevention and Management, University of British Columbia, Kelowna, British Columbia, Canada
<sup>5</sup>School of Health and Exercise Sciences, University of British Columbia, Kelowna, British Columbia, Canada
<sup>6</sup>International Collaboration on Repair Discoveries (ICORD), University of British Columbia, Vancouver, British Columbia, Canada
<sup>7</sup>Department of Economics, McMaster University, Hamilton, Ontario, Canada

**Acknowledgements** We would like to acknowledge Ali Shamaee for their support with study conceptualisation. We would also like to thank our study advisory panel consisting of young adults with the lived experience of a disability. We also like to acknowledge the support of our community partners including Canadian Council on Rehabilitation and Work, Abilities Centre, and National Educational Association of Disabled Students.

**Contributors** AJ: conceptualisation, methodology, writing - original draft, funding acquisition. KN: conceptualisation, methodology, writing - original draft. DVE: conceptualisation, methodology, writing - review and editing. MAMG: conceptualisation, methodology, writing - review and editing. KAMG: conceptualisation, methodology. ET: conceptualisation, methodology, writing - review and editing.

**Funding** This work was supported by a research operating grant from Accessibility Standards Canada, Employment and Social Development Canada, Government of Canada (#017323700).

**Competing interests** None declared.

**Patient and public involvement** Patients and/or the public were involved in the design, or conduct, or reporting, or dissemination plans of this research. Refer to the Methods section for further details.

**Patient consent for publication** Not applicable.

**Provenance and peer review** Not commissioned; externally peer reviewed.

**ORCID iDs**
Arif Jetha http://orcid.org/0000-0003-0322-7027
Dwayne Van Eerd http://orcid.org/0000-0002-5672-0168
Monique A M Gignac http://orcid.org/0000-0003-4445-3274
Kathleen A Martin Ginis http://orcid.org/0000-0002-7076-3594

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
