## [Reviewer comments · BMJ Open]

ARTICLE DETAILS

TITLE (PROVISIONAL)	Inclusion of young people with disabilities in the future of work: Forecasting workplace, labor market and community-based strategies through an online and accessible Delphi survey protocol
AUTHORS	Jetha, Arif; Nasir, Kay; Van Eerd, Dwayne; Gignac, M; Ginis, Kathleen; Tompa, Emile

VERSION 1 – REVIEW

REVIEWER	Jenni Ervasti Finnish Institute of Occupational Health, Topeliuksenkatu 41 a A, FI-00250, Helsinki, Finland
REVIEW RETURNED	17-Aug-2021

GENERAL COMMENTS	Inclusion in the future of work (study protocol) While I was expecting an intervention or a trial, this nevertheless is an interesting study protocol. The authors could shortly justify, why publishing of a study protocol is needed when conducting observational (as opposed to experimental) research? Title: Add “study protocol” to the title. Abstract: Here also, you should clearly state that this is a study protocol. Background: Replace “protocol will be implemented” with “study will be implemented” (or conducted). Protocol is a paper where you describe the study that you will then conduct. Please check these phrases throughout the text. What do you mean by: “Our study protocol has three specific objectives”? Do you mean that the planned study that is being described here has three objectives? Methods: Qualitative analysis is outside my area of expertise, so I hope another reviewer will evaluate methods and the planned Delphi study design. However, here also, replace phrases like “protocol will be implemented” with “study will be implemented”. The dates of the study are not clearly stated.
--

REVIEWER	Susan Peters Harvard Center for Work, Health and Wellbeing
REVIEW RETURNED	04-Oct-2021

GENERAL COMMENTS	Thank you for asking me to review this study protocol. This is a very timely study which will have important implications for the employment landscape for young people with disabilities. Introduction: 1. Can you include a sentence or two on how similar the Canadian context is to other countries (Lines 31-51)? If data are available,
---

providing statistics for other countries will enable the readers to understand the importance of this study in relation to their own context.

2. COVID-19 has impacted/accelerated trends related to the future of work, which will undoubtedly have impacts for new workers with disabilities. These trends will also have lasting effects long after the pandemic crisis is over. The accelerated timeline for many of the future of work changes (e.g., automation, use of technology to enable remote work) means that the findings and recommendations from your study are very relevant and timely. I would consider adding a sentence addressing this to the introduction.

Methods:

1. I appreciate the context section. Thank you for including this. Can you include a sentence up front that places this context more broadly and explaining why context matters. For example, "The social, political, and economic environment influences public and organizational policies that impact the nature of work and working conditions." You might want to reference this paper that describes the importance of these external drivers on workers' health and well-being: Sorensen, G., Dennerlein, J. T., Peters, S. E., Sabbath, E. L., Kelly, E. L., & Wagner, G. R. (2021). The future of research on work, safety, health and wellbeing: A guiding conceptual framework. *Social Science & Medicine*, 269, 113593.

2. Is there any scope through the Delphi methods to identify future of work trends that may be specific to this worker population (i.e., not identified through the formative systematic review)? I note that you will randomly allocate four of the nine (already identified) future of work themes to the participants. Is it possible there may be more unanticipated themes generated in the first round that would be important to explore outside of these four themes? Could that be addressed in the later round?

3. Can you explain your coding and analysis theoretical framework/approach? Will you be using a specific approach or conceptual framework to guide the coding process?

4. Page 15 Line 3: "structured online discussions" – this sounds like it could be a videoconference or similar, which I don't think is the case. Will there be opportunities for discussion? Or is it more likely that they participants will just be completing an online survey with open-ended questions? If there is a discussive component, this should be included in the methods.

5. There is six months between the two Delphi rounds. Could priorities and perspectives around the future of work change during this time, especially considering the acceleration of many of the future of work themes that you will be exploring? How might you address this?

6. Page 18, Line 10: Typo: "exist" should be "exists".

7. Page 18, Line 24: What is the process and how does your study methods change if you do not have consensus?

8. It is a real strength of this study to have a stakeholder team informing the design of the study and recruitment. It seems like a missed opportunity to not have these stakeholders informed in the analysis/interpretation of the study findings. This can also increase the trustworthiness of the data analysis process and help inform recommendations.

Discussion:

1. I would address (maybe in the limitations section) how the findings from this Canadian study are generalizable (or not) to other

	countries. 2. How could the findings from this study inform public and organizational policy and practice?
--	--

VERSION 1 – AUTHOR RESPONSE

REVIEWER 1

1. While I was expecting an intervention or a trial, this nevertheless is an interesting study protocol. The authors could shortly justify, why publishing of a study protocol is needed when conducting observational (as opposed to experimental) research?

We certainly appreciate the reviewer’s interest in our study protocol. The future of work is a constantly evolving topic. Past scholarship on the future of work has tended underscored the potential for dramatic shifts to the nature and availability of work. Based on past research, academics, labor market professionals and policy leaders focusing on the future of work have indicated the importance of generating strategies that can protect vulnerable workers, including people living with disabilities, from disadvantages related to the changing nature of work. We think our study protocol provides a rigorous approach to develop strategies that can be implemented in the present and can support workers with disabilities in the future of work. To address the reviewer’s comments, we have adapted the background section (Page 8, Paragraph 2):

“There is a need to design strategies that can ensure that the most vulnerable workers are protected from adverse changes in the future of work and are able to access potential opportunities. It is important to describe and implement a study protocol that can collect insights of diverse experts to forecast strategies that can be implemented in the present and ensure sustained employment of young people living with disabilities in the future.”

2. Add “study protocol” to the title

We have added study protocol to the title. Thanks for the suggestion.

3. Abstract: Here also, you should clearly state that this is a study protocol.

We have clearly stated that we are describing a study protocol in the abstract.

4. Background: Replace “protocol will be implemented” with “study will be implemented” (or conducted). Protocol is a paper where you describe the study that you will then conduct. Please check these phrases throughout the text.

We have adapted the text in the background and throughout the manuscript to address the reviewer’s comment.

5. What do you mean by: “Our study protocol has three specific objectives”? Do you mean that the planned study that is being described here has three objectives.

Apologies. We had intended to describe the planned study objectives. We have adapted this section for greater clarity.

6. Qualitative analysis is outside my area of expertise, so I hope another reviewer will evaluate methods and the planned Delphi study design. However, here also, replace phrases like “protocol will be implemented” with “study will be implemented”.

We have adapted the text throughout to reflect the reviewer’s feedback and appreciate the reviewer’s suggestion.

7. The dates of the study are not clearly stated.

The first round of the Delphi survey took place June-September 2021 (after this protocol was submitted for peer review). The second round of the survey will take place between March-June 2022. We have included these dates in description of the Delphi survey design (pages 12-13).

REVIEWER 2

8. Can you include a sentence or two on how similar the Canadian context is to other countries (Lines 31-51)? If data are available, providing statistics for other countries will enable the readers to understand the importance of this study in relation to their own context.

We appreciate the reviewer’s suggestion. The Canadian context is similar to other industrialized economies, especially those that are signatories to the United Nations Convention on the Rights of Persons with Disabilities. We taken steps throughout the manuscript to draw parallels between the Canadian context and other advanced economies.

See page 6, paragraph 2:

“The barriers to transitioning to employment reported by young Canadians with disabilities parallel those in other countries including the Australia, Netherlands and United Kingdom.”

See page 10, paragraph 1:

“Of note, findings from our study have implications for other industrialized countries who are also signatories to the CRPD.”

See page 19, paragraph 1:

“It is important to note that while our study is conducted within the context of Canada’s labor market and social policy context, it will produce insights that are relevant for the inclusion of people living with disabilities in other industrialized countries, especially those that are signatories of the CRPD.”

9. COVID-19 has impacted/accelerated trends related to the future of work, which will undoubtedly have impacts for new workers with disabilities. These trends will also have lasting effects long after the pandemic crisis is over. The accelerated timeline for many of the future of work changes (e.g., automation, use of technology to enable remote work) means that the findings and recommendations from your study are very relevant and timely. I would consider adding a sentence addressing this to the introduction.

We agree with the reviewer’s suggestion and have added the following to page 7, paragraph 2:

“The COVID-19 pandemic has accelerated employer investment into AI applications and other digital technologies to maintain productivity amidst lockdown and to address safety concerns related to disease transmission”

10. I appreciate the context section. Thank you for including this. Can you include a sentence up front that places this context more broadly and explaining why context matters. For example, “The social, political, and economic environment influences public and organizational policies that impact the nature of work and working conditions.” You might want to reference this paper that describes the importance of these external drivers on workers’ health and well-being: Sorensen, et al., 2021.

We think that it is important to highlight the importance of context with regards to supporting workers with disabilities in the future of work. To address the reviewer’s suggestion. We have adapted the methods and analysis section and included the following (page 9, paragraph 2):

“The sociopolitical and economic context plays a significant role in the determining the challenges and opportunities in the future of work as well as the types of labor market policies or employment programs that can be implemented to support young people living with disabilities”

11. Is there any scope through the Delphi methods to identify future of work trends that may be specific to this worker population (i.e., not identified through the formative systematic review)? I note that you will randomly allocate four of the nine (already identified) future of work themes to the participants. Is it possible there may be more unanticipated themes generated in the first round that would be important to explore outside of these four themes? Could that be addressed in the later round?

In the first round of the Delphi we ask an open-ended question asking participants about any additional reflections of the future of work and how it could impact young people with disabilities. These reflections will be synthesized in the second round of the Delphi. We have adapted the text to describe this question (page 13, paragraph 2):

“We also include a final open ended question for participants to include any additional reflections on the future of work and its implications for young people living with disabilities.”

12. Can you explain your coding and analysis theoretical framework/approach? Will you be using a specific approach or conceptual framework to guide the coding process?

Our study does not rely on a specific framework to inform data analysis. Instead, we take a constructivist perspective to inductively examine themes that emerge from the data. We have elaborated on our analytical approach on page 14, paragraph 1:

“Specifically, we take a constructivist perspective to inductively examine themes that emerge from the data to build an understanding of the specific strategies that can promote employment engagement in the future of work for young adults with disabilities. To implement our analytical approach, open-ended survey responses will be reviewed by members of the research team, and a codebook will be developed through several conversations.”

Also, on page 14, paragraph 1, we now highlight that findings will be member checked by our advisory panel to ensure trustworthiness of findings. We hope these additions provide the reviewer with additional insight into our analytical approach.

13. Page 15 Line 3: “structured online discussions” – this sounds like it could be a videoconference or similar, which I don’t think is the case. Will there be opportunities for discussion? Or is it more likely that they participants will just be completing an online survey with open-ended questions? If there is a discussive component, this should be included in the methods.

We apologize for the confusion. All participants will be completing multiple surveys with open-ended response options. There is no virtual conversation taking place in real-time. We have adapted the language highlighted by the reviewer for added clarity.

14. There is six months between the two Delphi rounds. Could priorities and perspectives around the future of work change during this time, especially considering the acceleration of many of the future of work themes that you will be exploring? How might you address this?

Indeed, priorities and working conditions could change between the two survey time points. We have accounted for changing perspectives on the future of work s using an open-ended question at the end of the survey. We have adapted the description of the survey on page 15, paragraph 1. We hope the additional description will address the reviewer’s feedback.

15. 6. Page 18, Line 10: Typo: “exist” should be “exists”.

We have made this edit and thank the reviewer.

16. 7. Page 18, Line 24: What is the process and how does your study methods change if you do not have consensus?

The reviewer raises an important concern that has been considered by members of the authorship team. There may be some strategies that are proposed where there is a lack of agreement. These strategies could still hold promise in promoting the employment of young people with disabilities. We will report on discordant strategies to encourage future research on their effectiveness. To elaborate on our methodological approach and address the reviewer’s feedback we have adapted our description of the survey (page 15, paragraph 2) and now note:

“We anticipate that there may be future of work strategies which do not achieve the threshold for consensus. These discordant strategies that may still be effective in promoting employment participation. We will document discordant strategies and share them in our final publications to encourage future research on their effectiveness.”

17. It is a real strength of this study to have a stakeholder team informing the design of the study and recruitment. It seems like a missed opportunity to not have these stakeholders informed in the analysis/interpretation of the study findings. This can also increase the trustworthiness of the data analysis process and help inform recommendations.

We certainly agree with the reviewer’s perspectives that our advisory panel is an important aspect of our research. Our intention is to include advisory panel members at all phases of the research including data analysis, development of study recommendation and knowledge mobilization. We have adapted our description of patient involvement on page 16, paragraph 1 and now note:

“The advisory panel will be asked to help interpret findings that emerge from the first and second rounds of the Delphi survey. Advisory panel members will also be engaged in the co-creation of study recommendation and knowledge mobilization outputs.”

18. I would address (maybe in the limitations section) how the findings from this Canadian study are generalizable (or not) to other countries.

We believe that findings will be relevant for other industrialized countries, especially those that are signatories of the United Nations Conventions on the Rights of Persons with Disabilities. We have added the following to our discussion section (page 19, paragraph 1):

“It is important to note that while our study is conducted within the context of Canada’s labor market and social policy context, it will produce insights that are relevant for the inclusion of

people living with disabilities in other industrialized countries, especially those that are signatories of the CRPD.”

19. How could the findings from this study inform public and organizational policy and practice?

Findings from our study will generate concrete strategies that can be implemented within organizational, educational, community-based or policy settings that can support the employment of young people with disabilities. Accordingly, our applied results can be integrated into the design of policies or practices. For added clarity we have adapted our discussion section (page 17, paragraph 2) and now note:

“Findings from our study can be used to update existing policies and program for people with disabilities or inform the design of innovative initiatives that are specifically relevant to the future of work.”

VERSION 2 – REVIEW

REVIEWER	Susan Peters Harvard Center for Work, Health and Wellbeing
REVIEW RETURNED	09-Dec-2021
GENERAL COMMENTS	The authors have addressed all of my concerns.